# An appropriate DNA input for bisulfite conversion reveals *LINE-1* and *Alu* hypermethylation in tissues and circulating cell-free DNA from cancers

**Trang Thi Quynh Tran[1,2], Tung The Pham[1], Than Thi Nguyen[1,3], Trang Hien Do[1], Phuong Thi Thu Luu[1], Uyen Quynh Nguyen[2], Linh Dieu Vuong[4], Quang Ngoc Nguyen[4], Son Van Ho[3], Hang Viet Dao[5], Tong Van Hoang[6], Lan Thi Thuong Vo[1,2]***

**1** Faculty of Biology, VNU University of Science, Vietnam National University, Hanoi, Vietnam, **2** VNU Institute of Microbiology and Biotechnology, Hanoi, Vietnam, **3** Department of Chemistry, 175 Hospital, Ho Chi Minh City, Vietnam, **4** Pathology and Molecular Biology Center, Vietnam National Cancer Hospital, Hanoi, Vietnam, **5** Endoscopic Centre, Hanoi Medical University Hospital, Hanoi, Vietnam, **6** Institute of Biomedicine and Pharmacy, Ha Dong, Vietnam

\* vothithuonglan@hus.edu.vn

## Abstract

The autonomous and active Long-Interspersed Element-1 (*LINE-1*, *L1*) and the non-autonomous *Alu* retrotransposon elements, contributing to 30% of the human genome, are the most abundant repeated sequences. With more than 90% of their sequences being methylated in normal cells, these elements undeniably contribute to the global DNA methylation level and constitute a major part of circulating-cell-free DNA (cfDNA). So far, the hypomethylation status of *LINE-1* and *Alu* in cellular and extracellular DNA has long been considered a prevailing hallmark of ageing-related diseases and cancer. This study demonstrated that errors in *LINE-1* and *Alu* methylation level measurements were caused by an excessive input quantity of genomic DNA used for bisulfite conversion. Using the minuscule DNA amount of 0.5 ng, much less than what has been used and recommended so far (500 ng-2 µg) or 1 µL of cfDNA extracted from 1 mL of blood, we revealed hypermethylation of *LINE-1* and *Alu* in 407 tumour samples of primary breast, colon and lung cancers when compared with the corresponding pair-matched adjacent normal tissue samples (P < 0.05–0.001), and in cfDNA from 296 samples of lung cancers as compared with 477 samples from healthy controls (P < 0.0001). More importantly, *LINE-1* hypermethylation in cfDNA is associated with healthy ageing. Our results have not only contributed to the standardized bisulfite-based protocols for DNA methylation assays, particularly in applications on repeated sequences but also provided another perspective for other repetitive sequences whose epigenetic properties may have crucial impacts on genome architecture and human health.

**Data Availability Statement:** All relevant data are within the manuscript and its Supporting Information files.

**Funding:** This study was funded by the Vingroup Innovation Foundation (VINIF) under project code VINIF.2022.DA00036. The funders had no role in study design, data collection and analysis, decision to publish, or preparation of the manuscript.

**Competing interests:** The authors have declared that no competing interests exist.

## Introduction

Nearly half of the human genome is comprised of retrotransposon elements, the repetitive sequences that can propagate via a "copy and paste" mechanism based on transcription and reverse transcription [1,2]. The autonomous and active Long-Interspersed Element-1 (*LINE-1*, *L1*) and the non-autonomous *Alu* elements are the most abundant repeated sequences in the human genome with an estimated more than $10^5$ and $10^6$ copies per genome, respectively [1,2]. Despite having a strong *cis* preference for its own RNA, *LINE-1* is also responsible for the *trans* mobilisation of *Alu* and other *SVAs* non-autonomous elements [3,4].

Recent emerging works have highlighted the fundamental role of *LINE-1* and *Alu* in hetero-chromatin compartmentalization, 3D genome folding, and overall genome spatial organization [5–7]. *LINE-1* can integrate into germ, embryonic and differentiated cells, providing a rich source of regulatory elements and functional modules, and resulting in a more diverse land-scape of somatic genome variation and phenotypic diversity [8,9]. Particularly, *Alu* sequences contain multiple potential splice donor/acceptor sites and binding sites for transcription factors such as enhancers and insulators [10], thus maintaining lineage-specific enhancer-promoter loops [11] and regulating transcriptional and post-transcriptional hubs [10–12].

*LINE-1* and *Alu* elements are heavily methylated and silenced in healthy controls but hypo-methylated and derepressed in cancer and ageing-related diseases [13,14]. In age-related dis-eases and cancers, *LINE-1* transposition promotes chromosomal rearrangements, transcriptional deregulation, oncogenic activation, and aberrations of non-coding RNAs [9,15]. In particular, *LINE-1* transcripts can cause heterochromatin erosion [16], induce the DNA damage response and cell cycle exit [17] and inflammation in ageing (inflammageing) [18]. Similarly, in diseases and cancer, *Alu* hypomethylation, corresponding to open chroma-tin, induces insertions and non-allelic homologous recombination, thus leading to instability and somatic complexity of the genome [19]. Hypomethylated *Alu*, behaving as cell-type-spe-cific enhancers for nearby genes [20], is strictly associated with an increase in *Alu* RNA tran-scribed by either RNA polymerase III (Pol III) or RNA polymerase II (Pol II), thus altering transcriptional and post-transcriptional hubs [10,21,22].

Contributing to 30% of the human genome, with 99% of their sequences being methylated [1], *LINE-1* and *Alu* elements constitute a major part of circulating-cell-free DNA (cfDNA) and undeniably contribute to the global DNA methylation level [14,23]. Moreover, the cfDNA integrity (cfDI) index, calculated as the ratio of long to short *Alu* fragments amplified at the same locus, is considered as a biomarker for cancer diagnosis [24]. cfDNA derived from cancer cells promotes tumour cell survival [25], facilitates colorectal tumour malignancy and stimulates the proliferation of hormone-receptor-positive breast cancer cells [26,27]. On the contrary, cfDNA modulates anti-tumour immunity in response to chemotherapy [28]. The methylated cfDNA fragments inhibit the proliferation of tumour cells [29]. Hence, fragmented cfDNA methylation signatures, associated with the dual effect of both pro- and anti-tumour, and global cellular DNA methylation can serve as a tool for real-time and accurate monitoring of cancer screening, accurate disease-progression surveillance and improvement of treatment [13,24].

DNA methylation at CpG sites at the 5'UTR of the human *LINE-1*, serving as an internal promoter, is one of the key mechanisms for silencing *LINE-1* [13]. Similarly, up to 33% of all CpG sites in the human genome are located in the *Alu* sequences, and the methylation of these sites is a primary mechanism in silencing almost all *Alu* elements in normal cells [14]. As a matter of fact, a hypomethylated status of the *LINE-1* and *Alu* in cellular DNA as well as in cfDNA has been considered hallmarks of ageing, age-related pathologies and cancer [19,30]. However, some recent studies have raised a challenging debate about whether *LINE-1* and *Alu* in cellular DNA and cfDNA is supposed to be hypomethylated or hypermethylated [31–33]. It

has been revealed that *LINE-1* hypermethylation and its silencing promoted the growth of tumour subclones [34] and metastasis [35], protected cancer cell subpopulations from lethal drug exposure [36], and allowed cancer cells to bypass antitumour immune responses [37]. Likewise, an increased level of *Alu* transcripts in diseases and cancers is associated with innate immune responses and apoptosis, protection of endogenous DNA damage and resistance to DNA damaging agents [38,39]. This proclivity towards the hypermethylated status of *LINE-1* and *Alu* in cancer cells, with their repression favouring fitness advantages for tumour survival and progression, suggests a re-evaluation of the prevailing hallmark of *LINE- 1* and *Alu* hypo-methylation in cancer.

Bisulfite conversion-based methods, the gold standard for quantifying and mapping methylation, with the ability to convert unmethylated cytosines to uracil residues while leaving methylated cytosines intact have been massively utilised to profile the methylation status of whole genomes and transposable elements through PCR-based amplification, post-PCR sequencing, and methylation arrays [40,41]. Using a wide range of DNA input (45 ng-1.6 μg) for bisulfite treatment, a significant conflict surrounding global methylation level, including that of *LINE-1* and *Alu*, in 32 reference samples, profiled either by whole genome bisulfite sequencing (WGBS) or pyrosequencing and methylation-specific PCR has been reported in a multicentre benchmarking study performed by 18 laboratories in 7 different countries [42]. Our previous studies have demonstrated that an excessive input of genomic DNA, corresponding to an excessive copy number of the *LINE-1* repeats used for bisulfite conversion caused errors in methylation measurements of *LINE-1* elements [43]. The effect of copy number variant on DNA methylation has been also mentioned using bisulfite-free methods such as methylated DNA immunoprecipitation (MeDIP) and methyl-CpG binding domain-based capture (MBDCap) [44].

In this study, we first focused on the quantitative assessment of the optimal DNA input amount for bisulfite conversions to eliminate the risk of incomplete conversions, and subsequently analysed the methylation profile of *LINE-1* and *Alu* in pair-matched tumour tissues and adjacent normal tissues of primary breast, lung and colon cancer samples. Since cfDNA retains the genetic and epigenetic characteristics of the tissue from which it was released [30], we also analysed the methylation profile of these elements in plasma from healthy individuals and lung cancer patients. As such, we demonstrated again the impact of DNA input in the methylation assessment of repetitive sequences. We established that these repeats were significantly hypermethylated in tumour tissue samples and cfDNA samples from lung cancer patients as compared with adjacent normal samples and cfDNA from healthy controls. More importantly, *LINE-1* hypermethylation in cfDNA is positively correlated with healthy ageing. Our findings provide information for a re-evaluation of the methylation profile of *LINE-1* and *Alu* and suggest another perspective for other repetitive sequences whose epigenetic properties may have crucial impacts on genome architecture and human health [45].

## Materials and methods

### Sample collection, DNA isolation and bisulfite conversion

Primary tumour tissue samples and their corresponding adjacent normal tissue samples were collected from fresh-frozen biopsies from 201 breast cancer patients, 133 colon cancer patients and 73 lung cancer patients. Sample classifications were done by pathologists at the Vietnam National Cancer Hospital and the 175 Hospital (Ho Chi Minh City). Blood samples were collected from 477 healthy participants and 296 patients with primary lung cancers. The recruitment period spanned from April 1st, 2023 to December 31st, 2023. Informed consent was obtained from healthy participants and patients in written form, and all collection methods

were performed in accordance with the relevant guidelines and regulations by the Ethics Committee of the Vietnam Academy of Science and Technology (01-2023/NCHG-HDDD). DNAs were extracted from tissues using the DNeasy Blood & Tissue Kit (Qiagen). After DNA quantification using the Qubit4 Fluorometer (Thermo Fisher Scientific), a concentration of 0.5 ng of genomic DNA was subjected to bisulfite conversion. One millilitre of blood was collected in K2-EDTA collection tubes, then centrifuged at 2000 g for 10 minutes at 4°C. Approximately 0.4 mL of plasma was transferred to a new tube and then centrifuged at 6000 g for 30 minutes at 4°C to remove residual blood and cell debris. cfDNA was extracted from the plasma using the MagMAX Cell-Free DNA Isolation kit (Thermo Fisher Scientific) and subjected to bisulfite conversion. All bisulfite reaction of DNA extracted from tissues and cfDNA was performed using the EZ DNA Methylation-Gold® kit (Zymo Research).

## Primer design

The *LINE-1* primer sets were designed to target the 5′ UTR region of the H1LS family (S1 Fig). The *Alu* primer sets targeted a region located in the 5' region of the *Alu* consensus sequences (S1 Fig). The methylation-dependent-specific PCR (MSP) primers used for profiling *LINE-1* and *Alu* methylation were derived from the CpG-containing sequences. The methylation-independent-specific PCR (MIP) primers were derived from the non-CpG-containing sequence [43] and used for the normalization of the DNA input. The specificity of the MSP primers, designed to specifically recognize methylated *LINE-1* and *Alu* sequences, was tested using (i) methylated human DNA and (ii) unmethylated human DNA (Zymo Research), both are treated by bisulfite and (iii) untreated human genomic DNA (Promega) as templates for qPCR. Amplification products were only obtained from the reactions with bisulfite-treated methylated human DNA, ensuring the accuracy of MSP primers designed for only the methylated targets. Primer sequences, amplicon lengths and qPCR conditions are presented in S1 Table.

## Determination of PCR amplification efficiency

A serial dilution of human methylated DNA amount (Zymo Research) ranging from 15 pg to 500 pg was bisulfite treated and used as templates for amplification either with the MIP primer set, specific to the bisulfite-converted reference sequence, or the MSP primer sets, specific to the methylated *LINE-1* and *Alu* sequences. The PCR amplification efficiency was assessed by (i) plotting the CT values against the DNA input and (ii) plotting the ΔCT values, calculated by subtracting the CT value of the MSP primer set from that of the MIP primer set against the DNA input.

## Quantification of bisulfite conversion efficiency

An artificially synthesized IC system, consisting of two sequences, named ConIC (Converted IC) and UnIC (Unconverted IC), containing both cytosine-free (CF) sequences and CpG-containing sequences, has been used in this study [43]. The ConIC and UnIC are both 122 bp in length, subcloned into the pTZ57R/T vector, and identical in sequence, except that all 15 cytosines in the UnIC are replaced by thymines in the ConIC, allowing the ConIC to be used as the calibrator for 100% bisulfite conversion efficiency, with the UnIC as the indicator to quantitatively evaluate the bisulfite conversion efficiency of the CpG sequence. The concentration of the linearized recombinant plasmids with the inserted sequences UnIC and ConIC was quantified using the Qubit™ 4 Fluorometer (Thermo Fisher Scientific) [43]. The copy number was estimated according to the molecular weight, amount, and length of the plasmid. To evaluate the impact of input amount on bisulfite conversions, a series of concentrations, equivalent to

$1010$–$10^6$ copies of the linearized plasmids UnIC and ConIC, was diluted in 500 ng of genomic DNA, converted by bisulfite, and subjected to qPCR with the CF and the MSP primer sets, respectively specific to the cytosine-free sequences and methylation-specific sequences of the IC [43]. In addition, a serial dilution of the human methylated DNA ranging from 500 ng to 0.5 ng was converted by bisulfite. Bisulfite-converted DNA inputs of 5 ng, 50 ng and 500 ng, diluted by 5, 50 and 500 folds and 20, 200 and 2000 folds, were respectively used for *LINE-1* and *Alu* methylation assessment. Two µL of the diluted solutions and the bisulfite-converted DNA inputs of 0.5 ng and 1 ng were then used as templates for qPCR.

## Quantification of cfDNA amount and cfDNA integrity (cfDI)

The total amount of plasma DNA was represented by either the *LINE-1* copy number or the *Alu* copy number present in plasma. To calculate *LINE-1* copy number, 1 µL of cfDNA was subjected to quantitative real-time PCR (qPCR) with the *LINE-1* primers corresponding to an 88 bp amplicon. To calculate cfDNA integrity, 1 µL of cfDNA was subjected to qPCR with the two *Alu* primer sets corresponding to 78 bp and 205 bp amplicons, respectively. Absolute quantification of *LINE-1* and *Alu* copy number was determined from a standard curve using 10-fold serial dilutions of human genomic DNA (Promega) ranging from 10.000 pg to 0.1 pg on the premise that a diploid genome (~6 pg) consists of $10^6$ copies of the *Alu* sequence and $10^5$ copies of the *LINE-1* sequence, respectively.

## Quantitative methylation-specific PCR (qMSP)

*LINE-1* and *Alu* methylation status was quantified via real-time PCR with a volume of 20 µL per reaction, using bisulfite-converted DNA as the template and the SsoAdvanced™ Universal SYBR® Green Supermix (Biorad). Real-time PCR assays were duplexed for both of the following reactions: (i) using the MIP primers to quantify bisulfite-converted reference sequences, and (ii) using the MSP primers to quantify methylated *LINE-1* and *Alu* sequences. Water with no DNA template was included in each PCR reaction as a control for contamination. All qPCR reactions were performed using the QuantStudio™ Real-time PCR instrument (Thermo Fisher Scientific).

## Methylation level calculation

The classical ΔΔCT approach using a calibrator was chosen to calculate the methylation level [46]. Validation of the ΔΔCT method when using a serial concentration of human methylated DNA that was treated with bisulfite and used as qPCR templates with the MIP and the MSP primer sets was analysed via the simple linear regression model. Three duplicated reactions were carried out for each sample: one using the MIP primer set to quantify the total DNA input (Ref) after bisulfite conversion and the others using the MSP primer sets to quantify the methylated *LINE-1* and *Alu* targets. A serial dilution of the human methylated DNA, treated with bisulfite, was used as the standard with a defined methylation level of 100%, and thus, as a calibrator for the measurement of *LINE-1* and *Alu* methylation levels. The relative level of methylated targets was calculated for each sample as follows: $\Delta\Delta CT_{Sample} = \Delta CT_{Sample} - \Delta CT_{Calibrator}$, where $\Delta CT_{Sample} = CT_{Sample/Methylated\ target} - CT_{Sample/Ref}$ and $\Delta CT_{Calibrator} = CT_{Calibrator/Methylated\ target} - CT_{Calibrator/Ref}$. $Methylation_{Sample} = 2^{-\Delta\Delta CT Sample} \times 100$ (%). Samples with both replicates returning a CT value greater than 29 were excluded from the study.

## Statistical analysis

In the box plot, the ΔCT values are represented as min to max with median values. In all dot plots, the ΔCT value, the methylation level of *LINE-1* and *Alu* are represented as median values with an interquartile range. All comparisons between more than two groups based on the quantitative values were assessed via the Kruskal-Wallis test. Comparisons between two groups based on the quantitative values were assessed via the Unpaired T test when normality was met, otherwise by the Mann-Whitney U test for independent samples and the Wilcoxon matched-pairs signed rank test for pair-matched samples. The relationship between the *LINE-1* and *Alu* methylation level and age was assessed via Spearman's rank correlation test. The linear relationship between the CT value and the genomic DNA input, and thus, the number of *LINE-1* and *Alu* copies input for qPCR, was evaluated through simple linear regression analysis. For all statistical analyses, a P-value of < 0.05 was considered significant. All analyses and graphing were performed using the GraphPad Prism® program version 9 (https://www.graphpad.com/scientific-software/prism/).

## Results

### Design of primers specific to *LINE-1* and *Alu* consensus sequences

The subfamily *LINE-1*, consisting of around one half million copies in the human genome, comprises almost all of the youngest and currently active human-specific *L1Hs* family [1,2]. Similarly, *Alu* elements have accumulated around 1 million copies in the human genome. Based on diagnostic nucleotides, *Alu* elements are classified into three major subfamilies: *AluJ* (the oldest), *AluS* (intermediate), and *AluY* (youngest) [47]. The most active human *Alu* subfamilies are *AluY*, which contribute to 13% of human-specific *Alu* elements [47]. Since these elements are highly heterogeneous, we first generated *LINE-1* and *Alu* consensus sequences based on a panel of young and old individual subfamily of each repeat element followed by Repeat-Masker and ClustalW algorithm [48]. The MSP primer sets were designed from these *LINE-1* and *Alu* consensus sequences and used for further methylation analysis (S1 Fig).

### Assessment of PCR amplification efficiency

A serial dilution of human methylated DNA amount ranging from 15 pg to 500 pg was bisulfite treated and used as templates in qPCR. The CT values have a strong linear relationship with the DNA input for the MIP primer set, specific to the bisulfite-converted reference sequence, and the MSP primer sets, specific to the methylated *LINE-1* and *Alu* sequences ($R^2$ > 0.99). There was a statistically insignificant difference (< 5%) in amplification efficiency (E value) with 94.58% for the reference (L1-Ref) amplicons, 91.92% for the methylated *LINE-1* (L1-Me) amplicons and 90.32% for the methylated *Alu* (Alu-Me) amplicons (Fig 1A and 1C). The difference in CT values of both the reference and the methylated target was then plotted against the logarithm of the template input amount, with absolute value of slopes of the fitted line to be 0.0145 and 0.0512 (P > 0.5), thus demonstrating the Livak formula suitable for the relative quantification of methylated *LINE-1* and *Alu* (Fig 1B and 1D). Taken together, these results indicate the designed MIP and MSP primer sets were optimised to PCR amplification, thus ensuring accurate measurement of *LINE-1* and *Alu* methylation levels using the Livak formula [46].

### The impact of excessive DNA amounts on methylation assessments

One of the technical weaknesses of bisulfite conversions is the incomplete conversion of unmethylated cytosines, leading to an overestimation of methylation level. On the other hand,

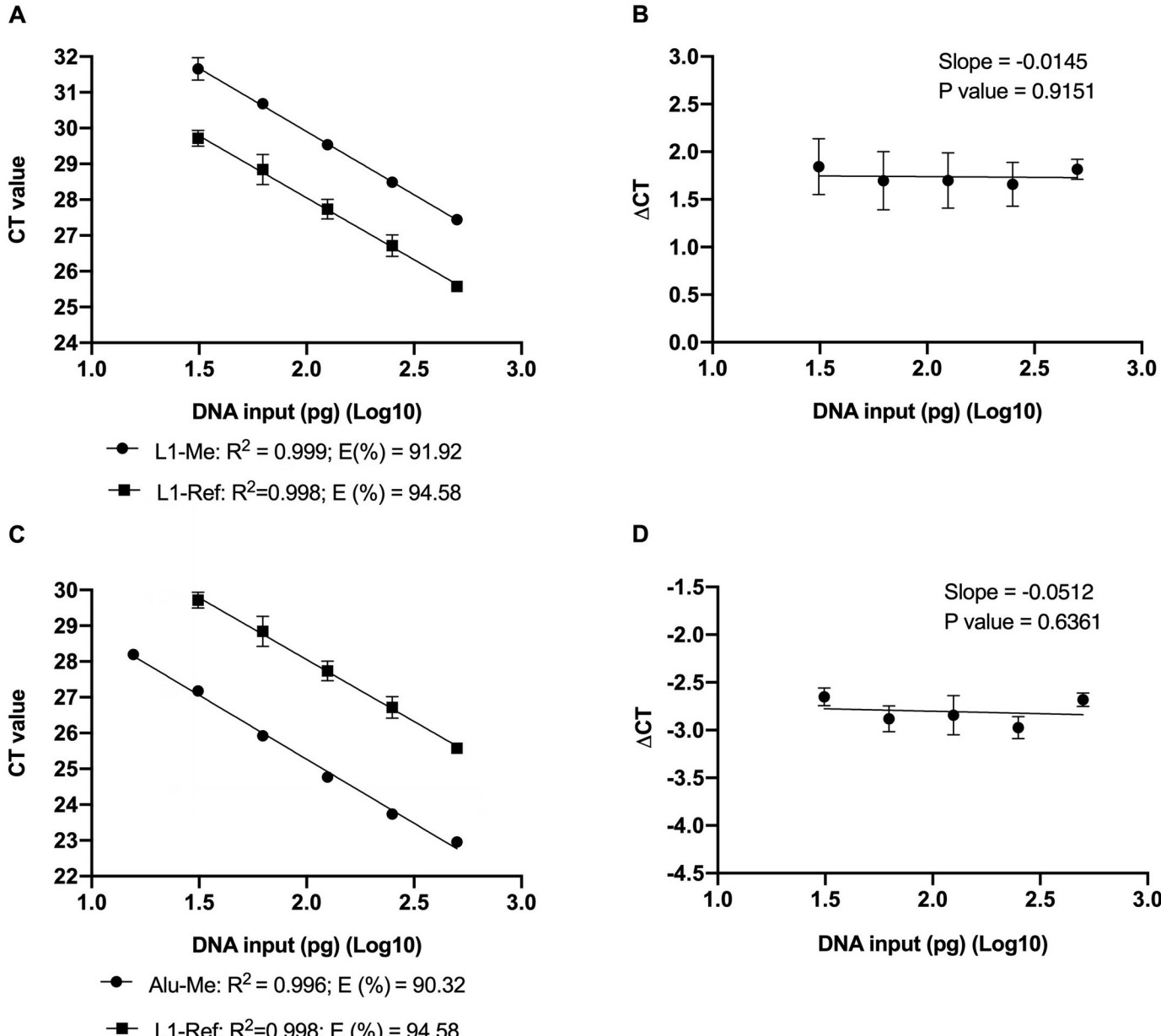

**Fig 1. Evaluation of PCR amplification efficiency of the MIP and MSP primer sets on the 100% human methylated DNA.** (**A, C**) A serial dilution of human methylated DNA ranging from 15 pg to 500 pg was bisulfite-treated and used as the template for qPCR with the MIP (L1-Ref) primers and the MSP (L1-Me and Alu-Me, respectively) primer sets specific to the *LINE-1* (A) and *Alu* (C) targets. The CT values have a strong linear relationship with the DNA input ($R^2 > 0.99$) and the amplification efficiency (E value > 90%) of the reference amplicon does not differ from that of the methylated amplicons (< 5% difference). (**B, D**) The ΔCT value, calculated by subtracting the CT value of the MSP primer set from that of the MIP primer set for each sample, is almost constant (P > 0.5), independent on DNA inputs, indicating the Livak ΔΔCT formula to be suitable for relative quantification of methylation level. Simple linear regression was used in statistical analysis. Number of observations for each assay was ≥ 3.

the conversion exclusively occurs on single-stranded DNA, thus, double-stranded DNA must be completely denatured during the bisulfite reaction [40,41]. Inefficient denaturation of excessive amounts of genomic DNA can result in an insufficient yield of the single-stranded DNA that is quantitatively measured through qMSP reactions, consequently leading to an underestimation of methylation level.

To understand the faulty underestimation of DNA methylation, an IC system, consisting of two artificial sequences, named ConIC and UnIC, containing both cytosine-free (CF) sequences and CpG-containing sequences, has been artificially synthesized [43]. They are identical in sequence, with the exception that in the ConIC, all cytosines have been converted into thymines. As such, the ConIC can be used as the calibrator for 100% bisulfite conversion efficiency, while the UnIC is the indicator to quantitatively evaluate the bisulfite conversion efficiency of the CpG sequence [43]. A serial dilution of the IC, corresponding to $10^{10}$, $10^9$, $10^8$, $10^7$ and $10^6$ copies, was spiked with 500 ng genomic DNA, converted by bisulfite, and used as templates for qMSP with the CF and MSP primer sets. Plotting the log of the UnIC copy number against the ΔCT value at each dilution shows significantly constant ΔCT values only within the range of $10^8$ to $10^6$ copies and starts to differ from the $10^9$ copies dilution. (Fig 2A). The $10^{10}$ copies of the UnIC gave a dramatical increase in the ΔCT value (Fig 2A). These phenomena were not observed with the ConIC, indicating that PCR amplifications were not inhibited by the excessive amount of template but instead by errors in CpG conversion due to an excessive amount of UnIC input for bisulfite.

The impact of the input DNA amount on methylation assessments was further investigated on *LINE-1* and *Alu* repeats in human methylated DNA (Zymo Research). Five different DNA quantities: 0.5 ng, 1 ng, 5 ng, 50 ng, and 500 ng of human methylated DNA were converted by bisulfite and subsequently used as templates for qMSP with the MIP and MSP primer sets specific to *LINE-1* and *Alu* sequences. All DNA sequences were methylated (100%) in this DNA, thus, theoretically, the ΔCT value should be constant regardless of the DNA amount for bisulfite conversions. Unexpectedly, the ΔCT values are constant only within the range of DNA input amounts from 0.5 ng to 5 ng with the *LINE-1* target and from 0.5 ng to 1 ng with the *Alu* target, and start to dramatically increase at DNA input amounts from 5 ng to 500 ng (Fig 2B and 2C). A higher input amount of the human methylated DNA correlated with a higher ΔCT value that corresponded to a lower methylation level (Fig 2D and 2E), indicating a faulty underestimation of *LINE-1* and *Alu* methylation, in other words, a faulty hypomethylation status of these repeat sequences.

Finally, the impact of different DNA amounts on *LINE-1* and *Alu* methylation assessments was investigated on DNA isolated from cancer samples. Bisulfite conversion was done with 3 different DNA quantities: 0.5 ng, 5 ng, and 50 ng on 30 matched pairs of DNA samples of breast tumour tissues and their corresponding adjacent tissues. As observed in Fig 2F and 2G, *LINE-1* and *Alu* methylation levels between breast tumours and their corresponding adjacent tissues significantly varied depending on the quantity of DNA input. Using 0.5 ng of genomic DNA, *LINE-1* and *Alu* are hypermethylated in breast tumour samples at 62.04% and 49.40% as compared to 40.81% and 35.32%, respectively ($P < 0.0001$ and 0.002) in adjacent tissue samples. Difference in methylation level at the 5 ng was only observed with *Alu* but not with *LINE-1*. However, when tested with 50 ng of DNA input, an insignificant difference in *LINE-1* and *Alu* methylation levels was seen between the tumour samples and the adjacent tissue samples (Fig 2F and 2G).

In addition, incomplete bisulfite conversion of the excessive DNA input was confirmed by using different DNA quantities ranging from 0.5 ng to 500 ng, on DNA samples of breast, colon and lung tumour tissues, which were converted by bisulfite and subsequently used as template for separate amplifications either with the primer sets specific to native consensus sequences or with the MSP primers specific to the methylated sequences. The native *LINE-1* product was successfully amplified from DNA input more than 0.5 ng; however, no PCR product was detected with DNA input of 0.5 ng (S2A Fig). Sequencing of the native *LINE-1* product with the native primers revealed the rare conversion of non-CpG cytosines, demonstrating renaturation to be a likely cause of incomplete conversion (S2B Fig). On the other hand, PCR

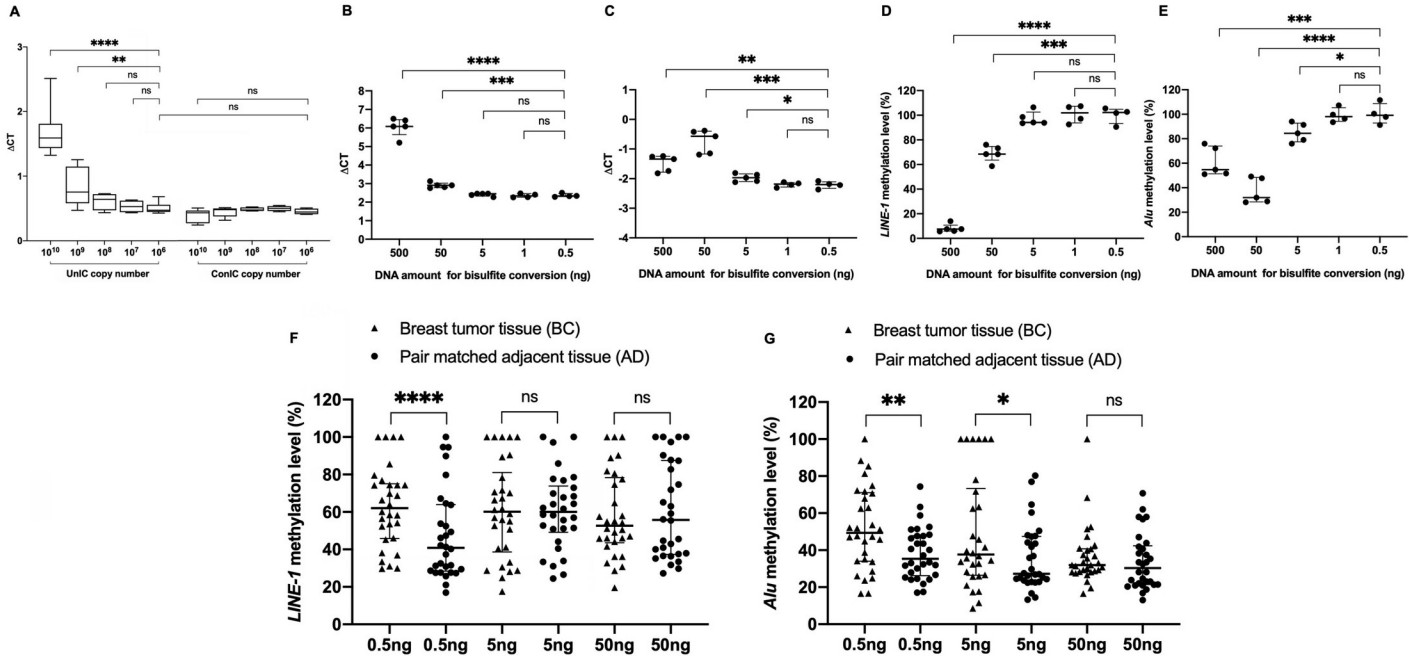

**Fig 2. The impact of excessive DNA amounts used for bisulfite conversion on the amplification efficiency and the methylation level of *LINE-1* and *Alu*.** (A) Correlation between the IC copy number (log10) and the ΔCT value. Serial dilutions of the UnIC and ConIC, corresponding to $10^{10}$, $10^9$, $10^8$, $10^7$ and $10^6$ copies, were spiked with 500 ng genomic DNA, bisulfite converted and amplified by two primer sets named CF and MSP [43]. The ΔCT ($CT_{MSP}$-$CT_{CF}$), calculated for each dilution, remained significantly constant only within the range of $10^8$ to $10^6$ copies but started to differ from the $10^9$ copies dilution. (B, C) The amplification efficiency of the reference template and the *LINE-1* and *Alu* methylated templates, on the 100% human methylated DNA. A serial dilution of human methylated DNA amounts, corresponding to 500 ng, 50 ng, 5 ng, 1 ng and 0.5 ng was treated by bisulfite. The ΔCT values significantly differ at input amounts from 500 ng to 0.5 ng. (D, E) An increase in methylation level of *LINE-1* (D) and *Alu* (E) was associated with a decrease in DNA input for bisulfite conversion. (F, G) Methylation level of *LINE-1* (F) and *Alu* (G) calculated from bisulfite-treated samples using 0.5 ng, 5 ng or 50 ng of genomic DNA extracted from breast tumours and their corresponding adjacent tissues. Hypermethylation of *LINE-1* and *Alu* elements in breast tumour samples was detected using 0.5 ng of DNA input solely. The Mann-Whitney U test and the Kruskal-Wallis test (A), the unpaired T-test (B-E) and the Wilcoxon matched-pairs signed rank test (F, G) were used for statistical analysis. (ns) nonsignificant; (*) P < 0.05; (**); P < 0.01; (***) P < 0.001; (****) P < 0.0001.

products were successfully obtained with DNA inputs of 0.5 ng and 50 ng using the MSP primers specific to the methylated *Alu*, indicating the presence of methylated *Alu* in bisulfite-converted DNA (S3A Fig). No PCR product was detected with the 0.5 ng-input DNA using the primer set specific to native *Alu* sequences, however, native *Alu* was successfully amplified from the 50 ng-input DNA (S3B Fig). This PCR product was then digested by the *Hpa*II restriction enzyme that recognized one CCGG site in the *Alu* sequences, indicating that some *Alu* sequences remained native even after bisulfite conversion (S3C Fig).

## Quantification of *LINE-1* and *Alu* copy number, and cfDNA integrity

The number of *LINE-1* copies in cfDNA from 180 healthy individuals (cfNC) and 213 lung cancer patients (cfLC) was measured in qPCR reactions to quantify the cfDNA concentration and manage the amount of cfDNA for the bisulfite reactions to eliminate the risk of incomplete conversions. On the other hand, the cfDNA, released from tumor cells, varies in size [24]. The ratio between long- (mostly > 200 bp), and short- (≤ 167 bp) *Alu* fragments, respectively, is referred to as DNA integrity (cfDI) index [24]. Hence, the number of *Alu* short (78 bp) and *Alu* long (205 bp) fragments in cfDNA was quantitatively calculated for 163 healthy individuals and 200 lung cancer patients. Absolute quantification of *LINE-1* and *Alu* number copies was done based on the premise that a diploid genome (~ 6 pg), respectively, consists of $10^5$ and $10^6$ copies of *LINE-1* and *Alu* sequences.

cfDNA was extracted from 0.4 mL plasma collected from one millilitre of blood. The number of *LINE-1* copies in one μL of the cfDNA from healthy participants was around $10^6$, and not significantly different from that of lung cancer patients (Fig 3A). cfDNA concentration in cancer is usually higher than that in healthy populations [24], however, in our study, this phenomenon was not yet observed, likely due to the primer set, designed to be specific to the 5-UTR regions of *LINE-1* which is truncated during transposition [15], hence, could not be an appropriate marker for evaluating an increase in cfDNA concentration. On the contrary, the number of short *Alu* copies in one μL of the cfDNA from healthy participants was less than that from lung cancer patients (P = 0.049); however, the number of long *Alu* copies from healthy participants was more than that from lung cancer patients (P = 0.022) (Fig 3B and 3C). Thus, the DNA integrity index (cfDI), calculated as the ratio of long to short *Alu* fragments, was higher in cfNC than that in cfLC (Fig 3D), indicating fragmented cfDNA increased in lung cancer patients, in line with cfDNA fragment sizes in cancer was shorter than that in healthy [24]. In addition, the number of *LINE-1* and *Alu* copies was also quantified in two DNA quantities of 0.5 ng and 5 ng from 60 tumour tissue and adjacent tissue samples of breast, colon and lung cancer. The number of *LINE-1* and *Alu* copies was proportional to the amount of DNA extracted from tissues; ranging from $10^6$ to $10^9$ copies, corresponding to 0.5 ng to 5 ng of genomic DNA, respectively (Fig 3E and 3F). Our previous study has indicated that a copy number of repeated sequences higher than $10^8$ can lead to incomplete bisulfite conversion [43, Fig 2A]. Hence, we chose 0.5 ng of genomic DNA isolated from tissue samples and 1 μL of cfDNA for further methylation analysis of these repeats on cancer tissues and cfDNA from lung cancer patients and healthy individuals. This DNA amount is the minimum recommended for bisulfite reaction by most manufacturers.

## Analysis of *LINE-1* and *Alu* methylation levels in tumour and adjacent tissues

Using 0.5 ng of genomic DNA for bisulfite conversion, the methylation level of *LINE-1* and *Alu* was quantitatively measured on 201, 133 and 73 pair samples of tumour tissue and their paired tumour-adjacent tissue corresponding to breast, colon and lung cancer. Significant hypermethylation was observed for *LINE-1* in tumour samples when compared with adjacent samples (Fig 4). As observed in Fig 4A, the methylated *LINE-1* level in tumour tissues of breast samples (52.11%), colon samples (38.67%) and lung samples (43.76%) were significantly higher than that in the adjacent normal tissue (46.27%, 34.89% and 42.78%, respectively) (P < 0.001–0.05). Similarly, significant *Alu* hypermethylation was revealed in breast (47.96%), colon (49.18%) and lung (66.12%) tumour samples as compared to the corresponding adjacent tissue samples (42.93%, 45.50% and 61.45%, respectively) (P <0.01–0.05) (Fig 4B). The large-scale number of breast and colorectal cancer samples reveals statistically significant association between *LINE-1* methylation and pathological stage (P = 0.0038), tumour size (P = 0.0131), as well as tumour grade (P < 0.0001) of breast cancer. However, there was insignificant association between the *LINE-1* methylation level and the clinicopathological characteristics of the colon cancer tissues, except for tumour stage (P = 0.0075). Likewise, there was insignificant association between the *Alu* methylation level and the clinicopathological characteristics of the breast and colon cancer tissues, except for pathological stage of the breast cancer (P = 0.0426) (S2 Table).

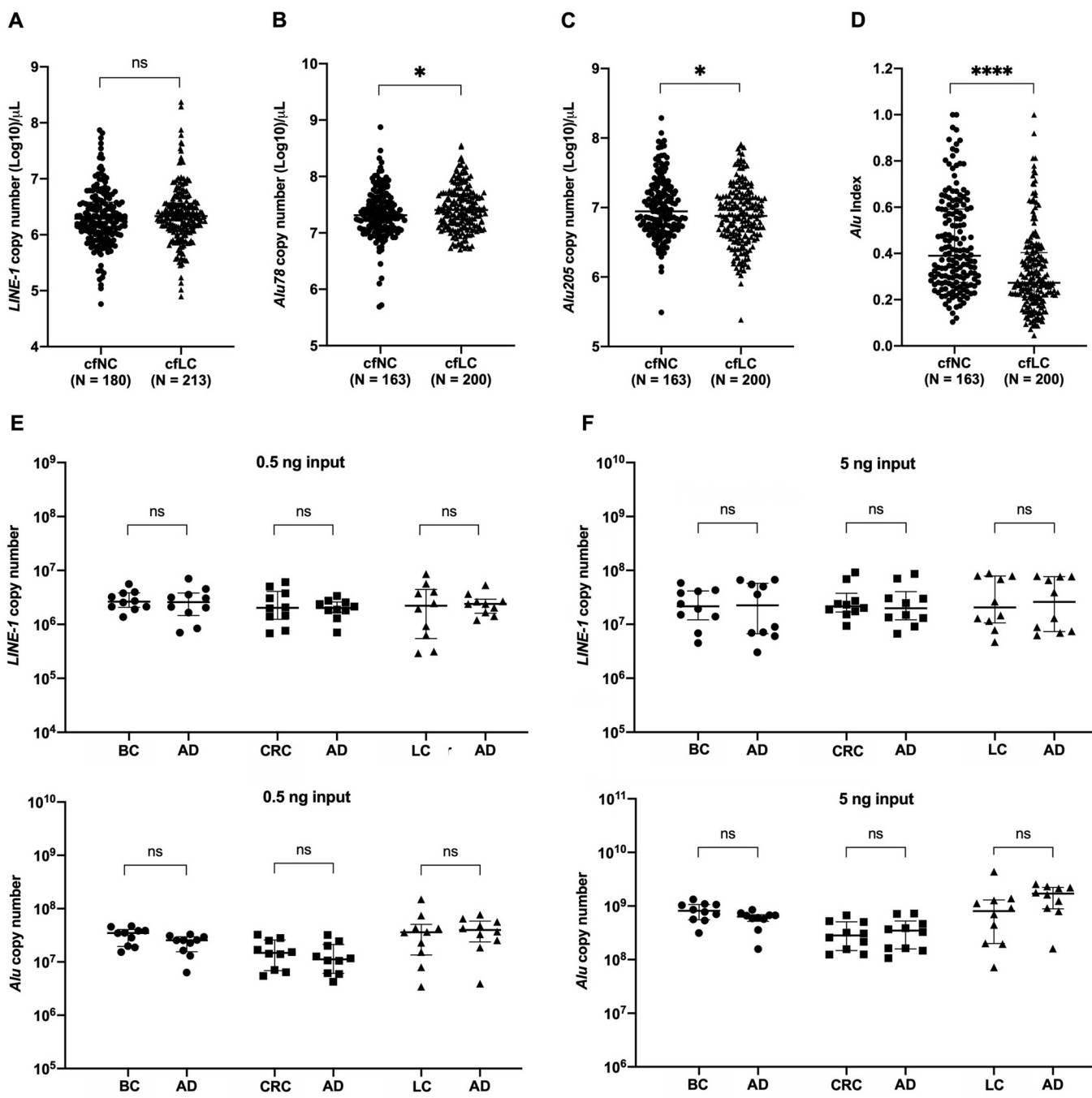

**Fig 3. Quantification of *LINE-1* and *Alu* copy number in cfDNA and cellular DNA.** The number of *LINE-1* copies (**A**), short *Alu* copies (**B**) and long *Alu* copies (**C**) in 1 μL cfDNA extracted from plasma collected from healthy controls (cfNC) and lung cancer patients (cfLC). The DNA integrity index (cfDI) was higher in cfNC than that in cfLC, indicating fragmented cfDNA increased in lung cancer patients (**D**). The number of *LINE-1* and *Alu* corresponding to different quantities of 0.5 ng (**E**) and 5 ng (**F**) of DNA extracted from adjacent tissue (AD) and tumour tissues samples from breast (BC), colon (CRC) and lung (LC) cancer. The number of *LINE-1* and *Alu* in 0.5 ng genomic DNA and in 1 μL of cfDNA is around $10^7$ copies, appropriating for bisulfite complete conversion. The Mann-Whitney U test (A-D), the Wilcoxon matched-pairs signed rank test (E, F) were used in statistical analysis. (ns) nonsignificant. (*) P < 0.05; (****) P < 0.0001.

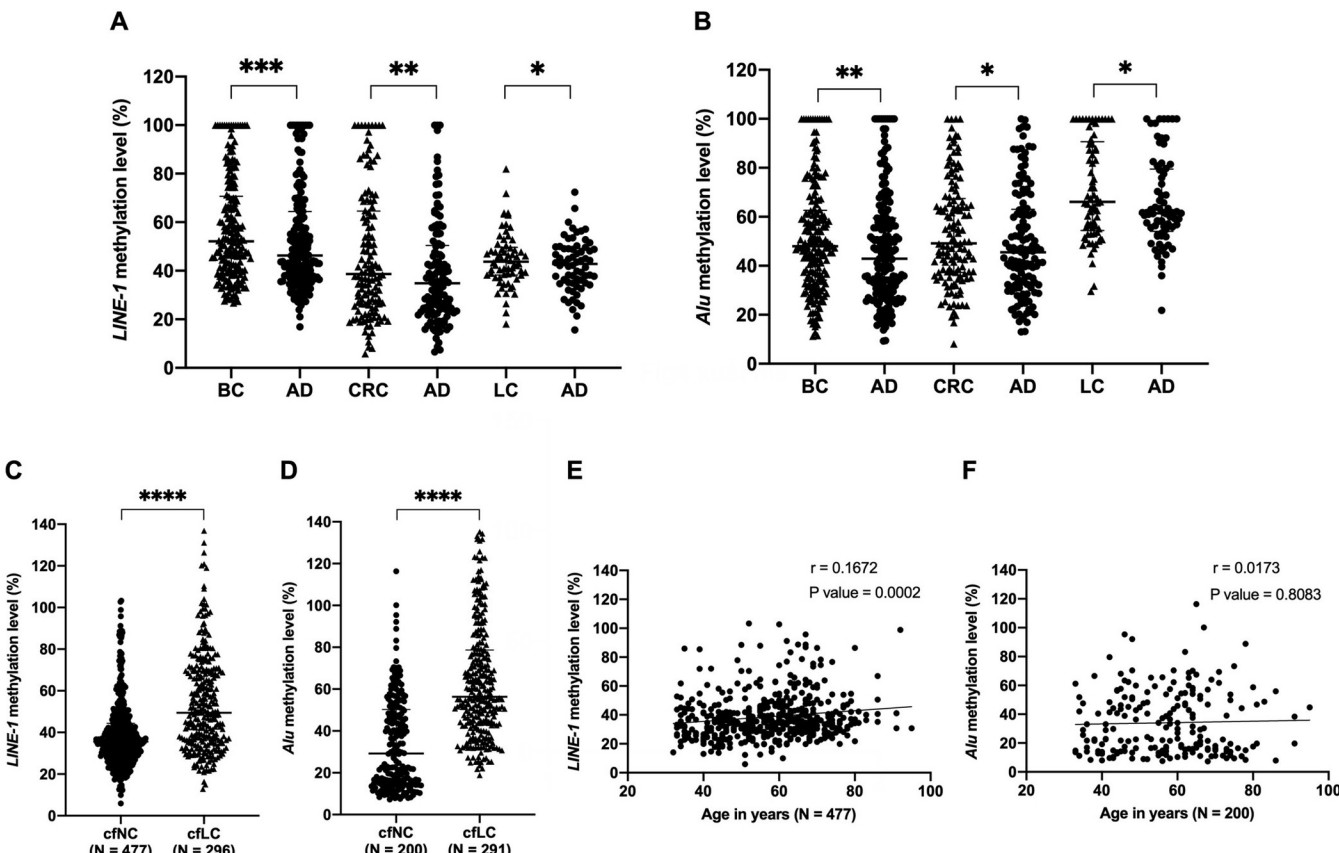

**Fig 4. *LINE-1* and *Alu* methylation in tissues and cfDNA from cancer and healthy individuals.** Hypermethylation of *LINE-1* (**A**) and *Alu* (**B**) in breast tumour (BC) samples, lung tumour (LC) samples and colon tumour (CRC) samples as compared with their corresponding adjacent tissue samples (AD). Methylation assessment was performed on DNA of 0.5 ng converted by bisulfite. Hypermethylation of *LINE-1* (**C**) and *Alu* (**D**) in cfDNA of lung cancer samples (cfLC) as compared to that in healthy samples (cfNC), respectively. Methylation level of *LINE-1* (**E**) but not of *Alu* (**F**) correlates with ageing in the healthy population. The Wilcoxon matched-pairs signed rank test (A, B), the Man-Whitney U test (C, D), and Spearman's rank correlation test (E, F) were used in statistical analysis. (*) P < 0.05; (**) P < 0.01; (***) P < 0.001; (****) P < 0.0001.

### Analysis of *LINE-1* and *Alu* methylation levels in the cfDNA of the healthy and lung cancers

Using one μL of cfDNA for bisulfite conversion, we analysed *LINE-1* and *Alu* methylation level in cfDNA samples extracted from 477 healthy donors and 296 primary lung cancer patients. Expectedly, significant hypermethylation was observed for *LINE-1* and *Alu* in cfDNA from cancer samples compared to that in healthy samples (Fig 4). The methylated *LINE-1* levels in lung cancer (49.43%) were significantly higher than that in the healthy samples (35.61%) (P < 0.0001) (Fig 4C). Likewise, significant hypermethylation was observed for *Alu* in cfDNA from lung cancer patients (56.31%) as compared to that from healthy donors (29.22%) (P < 0.0001) (Fig 4D). Unexpectedly, methylation level of *LINE-1*, but not of *Alu*, correlates with ageing as *LINE-1* is more methylated in older healthy individuals (P = 0.0002) (Fig 4E and 4F). This correlation was not observed in cancer populations (S4 Fig). In addition, there was a significant association between the pathological stage and metastasis of lung cancer patients and the *LINE-1* methylation in cfDNA solely (P = 0.0185) (S3 Table).

## Correlation between *LINE-1* and *Alu* fragmentation sizes and their methylation level in cfDNA

As cfDNA fragmentation is impacted by DNA methylation [49], we assessed the correlation between the copy number of *LINE-1*, short and long *Alu* sequences with their methylation level in cfDNA (S5 Fig). The number of *LINE-1* and short *Alu* copies, represented by amplicons of 88 bp and 78 bp in size and classified into short-size cfDNA proportion [27,49], correlated insignificantly with their methylation in healthy individuals and lung cancer patients. *LINE-1* and *Alu* elements are enriched by methylated CpG sites, which affect the preference of nucleases [27,50]. Thus, it is reasonable to speculate that this proportion of short-size methylated cfDNA fragments, having undergone fragmentation, became unrelated to their methylation status in both cfNC and cfLC. On the contrary, a decrease in long *Alu* copy number correlated with an increased *Alu* methylation level only in healthy individuals, which aligns with the negative relationship between cfDNA methylation and cfDNA fragment size [49]. It is worth noting that global nucleosome positioning is intrinsically affected by *Alu* elements [51] whose methylated status impacts nucleosome accessibility [49], explaining the no correlation between the copy number of long-size *Alu* and the increased *Alu* methylation in lung cancer. Moreover, the negative correlation between the *Alu* Index, typically used to measure cfDNA fragmentation, and *Alu* methylation level in both cfNC and cfLC (S5 Fig) also indicates a regulator role of DNA methylation in cfDNA fragmentation in both healthy individuals and cancer patients [49,50].

## Discussion

Alterations of CpG methylation status at either genome-wide or gene-specific levels have been confirmed as powerful biomarkers for diagnosis, prognosis, and prediction of diseases [13,24]. Bisulfite conversion-based methods are regarded as the gold standard for quantifying and mapping methylation from selected regions (MethyLight, Methylation-Sensitive High-Resolution Sanger sequencing, pyrosequencing) to whole genome (WGBS, EPIC Capture) [40,41]. A technical weakness of bisulfite-based methods is the incomplete conversion of unmethylated cytosines, thus producing errors in methylation measurement [52]. Actually, to verify bisulfite conversion efficiency in WGBS, methylated pUC19 and unmethylated lambda phage DNA are added to genomic DNA. However, these control DNA are used in smaller amounts (0.0005% and 0.01%, respectively, both equivalent to less than $10^8$ copies) that can be completely converted by bisulfite and thus, not be appropriate controls for the conversion of repeated targets [53]. High-throughput techniques such as Single-molecule real-time (SMRT) or Nanopore sequencing (ONT-seq), all of which are bisulfite-free methods but challenging due to the discordant data produced on the methylation profile of repeated sequences, too rich- or too poor-CpG sequences, and even different performances of DNA methylation calling tools is still controversial [41]. Until now, bisulfite-based methods have been widely used in many studies and commercial kits for DNA methylation analysis [40,54].

Bisulfite conversion exclusively occurs on single-stranded DNA, and is strictly dependent on DNA concentration, sequence complexity, GC content, and secondary structure elements [40,52]. Different commercial bisulfite conversion kits have been comprehensively evaluated for their bisulfite conversion efficiency [54]. These evaluations, however, investigated either endogenously single-locus targets or a small amount of artificially targeted templates but did not consider endogenous repeats or excessive amounts of artificial targets. Spiking an artificial unmethylated DNA molecule with genomic DNA revealed that complete bisulfite conversion is directly impacted by the copy number of the target, regardless of the genomic DNA amount (Fig 2A). Similarly, using an excessive amount of genomic DNA, corresponding to an

excessive number of *LINE-1* and *Alu* copies, for bisulfite conversion resulted in a contradicting increase of the ΔCT value in qPCR assays that eminently indicated false hypomethylation of these repeats (Fig 2B–2E). Incomplete conversion was revealed by successful amplification with the primers specific to the native consensus sequences from bisulfite-treated DNA input amounts more than 5 ng (S2 and S3 Figs). It is worth noting that 1 μg of genomic DNA input for methylation analysis, as usually recommended by most manufacturers, is equivalent to $10^6$ copies of single-locus targets, with only 2 copies per diploid genome; however, it would be equivalent to $10^{11}$–$10^{12}$ copies of *LINE-1* and *Alu* repeats, with up to $10^5$–$10^6$ copies per genome. It should be reminded that complete bisulfite conversions require the DNA input to consist of strictly single-stranded DNA, thus, an increase in the copy number of the double-stranded target could favourably promote renaturation, resulting in a decrease in the single-strand DNA yielded for bisulfite conversion. This phenomenon is similar to that in PCR reactions in which product yield would dramatically decline when the DNA template is over the threshold. Thus, it is reasonable that detrimental errors in *LINE-1* and *Alu* methylation assessment are caused by an excessive amount of DNA input for bisulfite conversion, the phenomenon has been previously described in our profiling of *LINE-1* methylation [43]. Based on obtained results, we speculate that the increase in copy number of *LINE-1* and *Alu* in cancer by their transposition [15], combined with a disproportionate amount of genomic DNA used for bisulfite conversion could cause an inefficient conversion by bisulfite and is likely to result in the faulty hypomethylation of these elements, which is detected in almost all reports on ageing and cancer described so far.

In this study, our findings indicated that a cellular DNA amount of less than 0.5 ng, or 1 μL of cfDNA extracted from 1 mL of blood was appropriate for methylation quantification of repeat sequences by bisulfite-based methods. We presented direct evidence of hypermethylation of *LINE-1* and *Alu* repeats, which is shown to predominantly occur in tissues as well as cfDNA from cancers (Fig 4). *LINE-1* and *Alu* are hypermethylated in breast, colon, and lung tumour tissues as compared to their adjacent normal tissues (Fig 4A and 4B). Similarly, they are also hypermethylated in cfDNA from lung cancer as compared to healthy controls (Fig 4C and 4D). These results are in direct contrast to the longstanding assumption of these repeats being hypomethylated in solid and liquid biopsy samples from diseases and cancers including breast, colon and lung cancer [13,14,30].

Hypomethylation of *LINE-1* and *Alu* repeats and their de-repression have been considered important features of ageing and cancer [9,13,19]. Recent emerging evidence, however, has revealed that this outdated assumption requires reconsidering [55]. In different types of cancer, *LINE-1* hypermethylation and its silencing promote the growth of tumour subclones and metastasis [34,35], allowing cancer cells to bypass antitumour immune responses and protecting cancer cell subpopulations from lethal drug exposure [36,37]. DNA-hypomethylation-induced overexpression of *LINE-1* repeats, responsible for the *trans* mobilisation of *Alu*, can result in detrimental impacts on the progression and survival of tumour cells [5]. On the other hand, an increased level of *Alu* transcripts in diseases and cancers has been explained by DNA hypomethylation and *vice versa*. This increase was, however, associated with the production of dsRNA, which can promote innate immune responses and apoptosis, prevent DNA damage and promote wound healing [20,38]. *Alu* siRNA processed from dsRNA can subsequently induce *Alu* methylation itself, elevating resistance to DNA-damaging agents [38,39]. Moreover, *Alu* RNA transcribed by Pol III can act as a modular *trans*-acting repressor of mRNA transcription through their binding on Pol II, and on protein translation; thus, inhibiting ribosome assembly [21,22]. Repression of *Alu*, mediated by SETDB1, an enzyme that deposits the H3K9me3 mark on *Alu* sequences, was observed in cancer cells [52]. These findings support our results that *LINE-1* and *Alu* are hypermethylated in cancer, which has been hidden due to

incomplete bisulfite conversions, and all together, providing a re-evaluation concerning the tumour-suppressive role of their methylation in cancer [55].

cfDNA retains the genetic and epigenetic characteristics of the tissue from which it was released [30]. As expected, hypermethylation of the *LINE-1* and *Alu*, observed in the lung tumour tissues, was also revealed in cfDNA from lung cancer patients as compared with healthy individuals (Fig 4C and 4D). Interestingly, *LINE-1* methylation in cfDNA significantly associated with pathological stage and metastasis in cancer patients (S3 Table). These elements abundant in cfDNA [23]; hence, their methylation assessment has become more favourable than other targets due to the sparse quantity of cfDNA. By demonstrating that a minuscule amount of cfDNA is preferable for their methylation assessment, our findings directly contributed to the standardized bisulfite-based protocols for DNA methylation assays, particularly in applications on repeated targets. The limitation of this study is the focus on the *LINE-1* and *Alu* elements as an entire group, instead of studying the methylation levels of a particular family or individual *LINE-1* and *Alu* loci, which are distinctly exhibited during normal development, ageing, diseases and cancer [2,47,56]. It is necessary for the locus- and tissue-specific patterns of their methylation to be investigated further in diseases and cancer. In addition, their hypermethylation in cfDNA, contributing to cfDNA methylation signatures can provide a promising tool for improving markers for cancer screening, accurate disease-progression surveillance and improvement of treatment [57,58].

## Conclusion

To summarize, this study has demonstrated that detrimental errors in measurement of the *LINE-1* and *Alu* methylation level are caused by an excessive input of genomic DNA, or an excessive copy number of these repeats, used for bisulfite conversion. A minuscule DNA amount, corresponding to a number of under $10^8$ copies was appropriate for bisulfite-based methods to identify their hypermethylation, which is shown to predominantly occur in tissues as well as cfDNA from cancers. Particularly, *LINE-1* and *Alu* methylation in cfDNA, associated with lung cancer, provides a potential DNA marker in real-time monitoring of diagnosis and prognosis in cancer. Our study suggests another perspective for other repetitive sequences whose epigenetic properties may have crucial impacts on genome architecture and human health.

## Supporting information

**S1 Table. Primer sets and quantitative real-time PCR conditions for quantification of copy number and measurement of methylation level of *LINE-1* and *Alu*.** The methylation-dependent-specific PCR (MSP) primers (Me-) are designed from the consensus sequences of *LINE-1* and *Alu* (S1 Fig). All non-CpG cytosines were replaced by 't" in the forward primers and by "a" in the reverse ones.
(PDF)

**S2 Table.** Association of clinicopathological characteristics of cancer patients with LINE-1 (A) and Alu (B) methylation status in tissues of breast cancer, colon cancer and lung cancer.
(PDF)

**S3 Table.** Association of clinicopathological characteristics of lung cancer patients with LINE-1 (A) and Alu (B) methylation status in cfDNA.
(PDF)

**S1 Fig. Identification of *LINE-1* and *Alu* consensus sequences followed by analyses on repeat masker [48] and ClustalW.** (A-C) Alignment of the 5' end region of *L1* elements. The

alignments include the old *L1PA* consensus sequence (A), the young *L1Hs* consensus representative to *L1-Ta-0* (B) and *L1-Ta-1* (C). The positions of the primers used in this study are shown with arrows, indicating the primers have been well designed to target all *L1Hs* elements including the *L1* consensus sequence (accession no. X58075). (D-F) Alignment of the *Alu* sequences of young (*AluY*), intermediate (*AluS*) and old (*Alu J*) individual *Alu* repeats. Based on these *Alu* consensus sequences, the primer sets specific to methylated *Alu* sequences were designed. The positions of the primers used in this study are shown with arrows.
(PDF)

**S2 Fig. Detection of the native *LINE-1* sequences after bisulfite conversion.** (**A**) The PCR product was successfully amplified with the primers specific to the native *LINE-1* from bisulfite-treated DNA with 5ng, 50ng, and 500 ng, on DNA samples of breast (1), colon (2) and lung tumour (3) tissues. However, no native *LINE-1* product was detected when the DNA input was 0.5 ng. (—): Negative control without DNA template. (**B**). Direct sequencing of PCR products amplified from bisulfite-treated DNA with an input amount of 0.5 ng using the MIP primers and an input amount of 50 ng using the primers specific to the native *LINE-1* sequences. Almost non-CpG cytosines remained in DNA input of 50 ng. Rare conversions of non-CpG cytosines were underlined.
(PDF)

**S3 Fig. Detection of the native *Alu* sequences after bisulfite conversion.** (**A**). The PCR product was successfully amplified with the MSP primers specific to the methylated *Alu* sequences from bisulfite-treated DNA with different DNA quantities of 0.5 ng and 50 ng on DNA samples of breast (1) and colon (2) tumour tissues. (+): Recombinant plasmid containing the methylated *Alu* sequences used as template. (**B**). The PCR product was successfully amplified with the primers specific to the native *Alu* sequences from bisulfite-treated DNA of 50 ng input. However, no native *Alu* product was detected when the DNA input was 0.5 ng. PC: Positive control with DNA extracted from a blood sample as template. NTC: Negative control without DNA template. (**C**). Digestion of the PCR products from (B) with the *Hpa*II restriction enzyme, recognizing one CCGG site in the *Alu* sequences, indicated that some native *Alu* sequences remained after bisulfite conversion of a DNA input of 50 ng.
(PDF)

**S4 Fig.** LINE-1 (A) and Alu (B) methylation levels in cfDNA do not correlate with ageing in lung cancer patients. Methylation assessments were performed on one microlitre of bisulfite-converted cfDNA. The Spearman's rank correlation test (A, B) was used in statistical analysis.
(PDF)

**S5 Fig. Correlation between *LINE-1* and *Alu* fragmentation sizes with their methylation level in cfDNA from healthy individuals (cfNC) and lung cancer patients (cfLC).** No relationship was observed between the copy numbers of short *LINE-1* (A) and short *Alu* (B) with their methylation level in both cfNC and cfLC. A negative correlation was detected between the long *Alu* copy number and *Alu* methylation in cfNC but not in cfLC (C). *Alu* Index negatively correlated with *Alu* methylation in both cfNC and cfLC (D). Correlations were assessed using Spearman's rank correlation test.
(PDF)

## Acknowledgments

The authors would like to thank Pham ATD at the Biomedical Lab, VNU University of Science for technical support.

## Author Contributions

**Conceptualization:** Lan Thi Thuong Vo.

**Data curation:** Trang Thi Quynh Tran, Tung The Pham, Than Thi Nguyen, Trang Hien Do, Phuong Thi Thu Luu.

**Formal analysis:** Trang Thi Quynh Tran, Tung The Pham, Than Thi Nguyen, Uyen Quynh Nguyen.

**Funding acquisition:** Lan Thi Thuong Vo.

**Methodology:** Trang Thi Quynh Tran, Tung The Pham.

**Resources:** Linh Dieu Vuong, Quang Ngoc Nguyen, Son Van Ho, Hang Viet Dao, Tong Van Hoang.

**Supervision:** Lan Thi Thuong Vo.

**Validation:** Trang Thi Quynh Tran, Than Thi Nguyen, Phuong Thi Thu Luu.

**Visualization:** Tung The Pham.

**Writing – original draft:** Trang Hien Do, Uyen Quynh Nguyen.

**Writing – review & editing:** Lan Thi Thuong Vo.

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
