## [Decision Letter · Decision Letter 0]

8 Nov 2024

PONE-D-24-37696An appropriate DNA input for bisulfite conversion reveals LINE-1 and Alu hypermethylation in tissues and circulating cell-free DNA from cancersPLOS ONE

Dear Dr. Vo,

Thank you for submitting your manuscript to PLOS ONE. After careful consideration, we feel that it has merit but does not fully meet PLOS ONE’s publication criteria as it currently stands. Therefore, we invite you to submit a revised version of the manuscript that addresses the points raised during the review process.

We look forward to receiving your revised manuscript.

Kind regards,

Osman El-Maarri, Ph.D

Academic Editor

PLOS ONE

**Journal Requirements:**

This study was funded by the Vingroup Innovation Foundation (VINIF) under project code VINIF.2022.DA00036

Reviewers' comments:

Reviewer's Responses to Questions

**Comments to the Author**

1. Is the manuscript technically sound, and do the data support the conclusions?

Reviewer #1: Yes

Reviewer #2: Yes

Reviewer #3: Yes

2. Has the statistical analysis been performed appropriately and rigorously? 

Reviewer #1: Yes

Reviewer #2: Yes

Reviewer #3: Yes

3. Have the authors made all data underlying the findings in their manuscript fully available?

Reviewer #1: Yes

Reviewer #2: Yes

Reviewer #3: Yes

4. Is the manuscript presented in an intelligible fashion and written in standard English?

Reviewer #1: Yes

Reviewer #2: Yes

Reviewer #3: Yes

5. Review Comments to the Author

**Reviewer #1:** I consider these data worth to be published. In spite it "contradicts" the widely accepted results regarding hypomethylation in repeated sequences, I think we, the scientific comnunity should try to replicate our epigenetic studies with a smaller amount of DNA. What I consider as very novel results are those from cfDNA, and I think authors have used good control groupd for comparison. I also think these data are worth to be publishes since they are supported by a previous study, also, other different authors had already proposed to explore new mechanistics rehardin Alu and LINE-1 in cancer development

**Reviewer #2:** This article is undoubtedly of interest not only to molecular biologists and also to oncologists. Article corresponds to the topics of this journal however there are several clarifying questions (comments) to the authors:

Questions:

1. Quantification of LINE-1, Alu copy number and measurement of methylation level was performed on tissue samples and cell-free DNA from lung cancer patients? Is there any results received on samples from breast cancer patients and colorectal cancer patients?

2. Why in this article did not discussed correlation (or absent of correlation) between LINE-1, Alu copy number and methylation level in tissue samples and cell-free DNA?

3. Authors quantified copy number and methylation level for the total pool of Alu elements (“short” and “long”?). This information did not indicated in results. Although in “Materials and methods” indicate that both sequences were determined.

**Reviewer #3:** The study aims to address the technical issue of the influence of DNA concentration on the efficiency of DNA conversion and to investigate the methylation level of repetitive sequences in the tumor.

The reviewer has several comments and observations.

Major

Page 7, line 170. Authors should provide information about the total size of the UnIC and ConIC plasmids, size of inserts and number of C in this inserts to understand the real quantitative differences between these plasmids for the bisulfite conversion. In ref. 44 there is only one indirect mention of the size of these plasmids (in S3 Table). "Linearized recombinant plasmids (4 pg equivalent to 10*6 copies)"

Page 11, line 287. What amount of DNA was used as template in the PCR reactions? Was it constant or did it vary with the amount of DNA taken for bisulfite conversion? This information should be provided.

In the discussion, the authors suggest that hypomethylation in tumor cells observed in many studies is a consequence of cytosine conversion to the methylation level of LINE-1 and discuss the mechanism of such an effect for MSP. However, hypomethylation of LINE in tumors has also been shown using direct sequencing (pyrosequencing, massive parallel sequencing), when incomplete cytosine conversion can be directly assessed in the final data. Authors should include a discussion of this issue in the Discussion section.

Minor

Page 2, line 44. Reference "221" should be replaced by "1"

Page 3, line 70. Reference is lost

Page 9, line 223-226. Two first sentences of the Result section seem to be similar to each other and should be re-written

Page 13, line 335. Change 'S2A Fig.' to 'Fig. S2A'

6. PLOS authors have the option to publish the peer review history of their article (what does this mean?). If published, this will include your full peer review and any attached files.

Reviewer #1: **Yes: **Octavio Jiménez-Garza

Reviewer #2: No

Reviewer #3: No

---

## [Author Response · Author response to Decision Letter 0]

14 Nov 2024

Vo Thi Thuong Lan, Assoc. Prof.

Faculty of Biology, VNU University of Science,

334, Nguyen Trai, Thanh Xuan

Ha Noi, Viet Nam.

 PLoS ONE Editorial Board

Ha Noi, December 14th, 2024 

Dear Members of the Editorial Board,

We are deeply grateful to the Reviewers for the supportive comments and suggestions that have allowed us to considerably improve our manuscript titled “An appropriate DNA input for bisulfite conversion reveals LINE-1 and Alu hypermethylation in tissues and circulating cell-free DNA from cancers” by Tran et al., submitted to the PLoS ONE under the reference ID: PONE-D-24-37696. After meticulous consideration of the comments from the Reviewers, we have now rewritten the manuscript to address their feedback and align more closely to the journal’s requirements. Modifications have been tracked in the revised manuscript and are detailed point by point in the following section.

We confirm that this study was funded by Vingroup Innovation Foundation (VINIF) under project code VINIF.2022.DA00036. The funders had no role in study design, data collection and analysis, decision to publish, or preparation of the manuscript, however, decided on the journal for publication.

In the resubmitted manuscript, we have:

Added one sub-section in the Results “Correlation between LINE-1 and Alu fragmentation sizes and their methylation level in cfDNA” and S5 Fig. 

Added a supplementary figure S5 Fig.

Added three citations [ref. 50 – ref. 52] to support the correlation between LINE-1 and Alu fragmentation sizes and their methylation level in cfDNA.

Added one citation [ref. 54] as documentation of DNA controls used in whole genome bisulfite sequencing (WGBS) to verify bisulfite conversion efficiency.

To better adhere to the journal requirements, we have:

Reformatted the manuscript to meet PloS ONE’s style requirements.

Stated the role of the funders in the study as follows: “The funders had no role in study design, data collection and analysis, decision to publish, or preparation of the manuscript, however, decided on the journal for publication.”

Confirmed the ethics statement in Materials and Method is correct and removed this statement from the section after the main text.

Confirmed the reference list is complete and correct.

We would like to note that author corrections were made for five papers. However, these corrections are minor, either not related to or do not alter experimental results, their interpretation or the conclusions of the articles, thus do not affect our citations.

Brief details of the corrections are as follows:

[Ref 15] corrected and added author affiliations. doi: 10.1038/s41588-023-01319-9

[Ref 18] corrected figures with data points corresponded to the means ±1 standard deviation, rather than actual data points, as well as files and data related to this error. doi: 10.1038/s41586-019-1350-9

[Ref 23] added missing Supplementary Tables. doi: 10.1093/bioinformatics/btae600

[Ref 45] corrected reversed figure legends and the text mentioning these figures. Genome Res. 2011 Jan;21(1):146

[Ref 50] corrected the molarity of the resulting solution in the methods section. doi: 10.1093/nar/gkp637

Reviewer #1: 

We are grateful to the Reviewer for your comment. We are currently investigating the methylation level of repetitive sequences in cancer tissues as well as in circulating cell-free DNA (cfDNA) from gynecologic, gastrointestinal cancer patients and healthy individuals.

Reviewer #2:

1. Quantification of LINE-1, Alu copy number and measurement of methylation level was performed on tissue samples and cell-free DNA from lung cancer patients? Are there any results received on samples from breast cancer patients and colorectal cancer patients?

In our study, the copy numbers of LINE-1 and Alu and their methylation levels were quantified in tissue samples from breast, colon and lung cancer patients (Figs 3E and F, Figs 4A and B). However, their copy numbers and methylation levels were quantified in cfDNA from only lung cancer patients (cfLC) and healthy individuals (cfNC) (Figs 3A-D, Figs 4C and D).

2. Why in this article did not discussed correlation (or absent of correlation) between LINE-1, Alu copy number and methylation level in tissue samples and cell-free DNA?

We are grateful to the Reviewer for your interesting comment. In this study, we assessed the correlation between the copy numbers of LINE-1 and Alu and the DNA input amount (ng) on tissue samples (Figs 3 E and F) in order to choose the appropriate amount of DNA input (0.5 ng of cellular DNA, corresponding to less than 108 copies of LINE-1 and Alu), for bisulfite reactions. Since the number of tissue samples used for copy number quantification (N = 60) was greatly less than that for methylation level quantification (N = 201, 134 and 73 matched pairs of tumor-normal tissues from breast, colon and lung cancer, respectively), the correlation between LINE-1 and Alu copy number and their methylation level in tissue samples were not mentioned.

On the other hand, LINE-1 and Alu sequences, predominant components of cfDNA, are overrepresented by past 200% compared to their composition in the genome [ref 24], thus, the copy numbers of LINE-1 and Alu in cfDNA from lung cancer patients (cfLC) and healthy individuals (cfNC) were carefully assessed with a large number of samples (N >150) (Figs 3 A-D). Following the Reviewer’s suggestion, we included that a decrease in long Alu copy number but not in that of short Alu correlated with an increased Alu methylation level cfNC, which is in line with the fact that shorter cfDNA correlated with an increased DNA methylation [1]. Moreover, the negative correlation between Alu methylation level and Alu Index, typically used to measure cfDNA fragmentation also indicates a regulator role of DNA methylation in cfDNA fragmentation in both healthy individuals and cancer patients [1,2]. In addition, there was no correlation between LINE-1 copy number and its methylation level in cfNC and cfLC.

In the revised manuscript, we have adjusted the Results section to include this assessment under the sub-section “Correlation between LINE-1 and Alu fragmentation sizes and their methylation level in cfDNA” and added a supplementary figure S5 Fig (line 451, page 18).

3. Authors quantified copy number and methylation level for the total pool of Alu elements (“short” and “long”?). This information did not indicate in results. Although in “Materials and methods” indicate that both sequences were determined.

We thank the Reviewer for this recommendation. As explained above, we have adjusted the Results section to include an assessment of correlation between the copy number of short Alu, long Alu, and the Alu Index with the total Alu methylation level in the revised manuscript (line 451, page 18, S5 Fig).

References

1. Kim M, Park J, Oh S, Jeong BH, Byun Y, Shin SH, et al. Deep learning model integrating cfDNA methylation and fragment size profiles for lung cancer diagnosis. Scientific Reports. 2024; 14(1):14797. doi: 10.1038/s41598-024-63411-2 PMID: 38926407

2. An Y, Zhao X, Zhang Z, Xia Z, Yang M, Ma L, et al. DNA methylation analysis explores the molecular basis of plasma cell-free DNA fragmentation. Nature Communications. 2023; 14:287. Doi: 10.1038/s41467-023-35959-6.

Reviewer #3: The study aims to address the technical issue of the influence of DNA concentration on the efficiency of DNA conversion and to investigate the methylation level of repetitive sequences in the tumor.

The reviewer has several comments and observations.

Major:

1. Page 7, line 170. Authors should provide information about the total size of the UnIC and ConIC plasmids, size of inserts and number of C in these inserts to understand the real quantitative differences between these plasmids for the bisulfite conversion. In ref. 44 there is only one indirect mention of the size of these plasmids (in S3 Table). "Linearized recombinant plasmids (4 pg equivalent to 10*6 copies)"

We appreciate the comment from the Reviewer. The ConIC and UnIC inserts are both 122 bp in length, subcloned into the 2886 bp pTZ57R/T vector, and were initially reported by Lan et al. (2019) [1]. It is crucial to note that these inserts are identical in sequence, except that all 15 cytosines in the UnIC are replaced by thymines in the ConIC, allowing the ConIC to be used as the calibrator for 100 % bisulfite conversion efficiency, with the UnIC as the indicator (Fig 1). We have revised the Materials and Methods section in our manuscript to include these IC definitions (line 168, page 7).

Fig 1. Nucleotide sequences (122 bp) of the UnIC and the ConIC fragments. All “C” (highlighted in red) in the UnIC sequence are replaced by ‘t” in the ConIC sequence. The CF-F/R primer set (highlighted in yellow) targeting the CF sequence in the IC is used to evaluate DNA recovery, and the MSP-F/R primer set (underlined in the UnIC) is used to assess bisulfite conversion efficiency [1].

2. Page 11, line 287. What amount of DNA was used as template in the PCR reactions? Was it constant or did it vary with the amount of DNA taken for bisulfite conversion? This information should be provided.

We thank the Reviewer for this recommendation. Five different DNA quantities of 0.5 ng, 1 ng, 5 ng, 50 ng, and 500 ng of human methylated DNA were converted by bisulfite. Bisulfite-converted DNA inputs of 5 ng, 50 ng and 500 ng, diluted by 5, 50 and 500 folds and 20, 200 and 2000 folds, were respectively used for LINE-1 and Alu methylation assessment. Two μL of the diluted solutions and the bisulfite-converted DNA inputs of 0.5 ng and 1 ng were then used as templates for qPCR. We have adjusted the Materials and Methods section of our manuscript to include this information (line 183, page 8).

3. In the discussion, the authors suggest that hypomethylation in tumor cells observed in many studies is a consequence of cytosine conversion to the methylation level of LINE-1 and discuss the mechanism of such an effect for MSP. However, hypomethylation of LINE in tumors has also been shown using direct sequencing (pyrosequencing, massive parallel sequencing), when incomplete cytosine conversion can be directly assessed in the final data. Authors should include a discussion of this issue in the Discussion section.

We fully agree with the Reviewer. The most common bisulfite-conversion-based techniques that have been performed to profile LINE-1 methylation are pyrosequencing and whole genome bisulfite sequencing (WGBS). To verify bisulfite conversion in WGBS, methylated pUC19 and unmethylated lambda phage DNA are added to genomic DNA. However, these control DNA are used in smaller amounts (0.0005% and 0.01%, respectively, both equivalent to less than 108 copies) that can be completely converted by bisulfite and thus, not be appropriate controls for the conversion of repeated targets [2,3]. In addition, the efficiency of non-CpG cytosine bisulfite conversions, used as built-in controls to verify bisulfite conversion in pyrosequencing can be hindered by sequence complexity, GC content, secondary structure elements and even a given cytosine in a particular sequence [4, ref. 5]. Based on our experiences, non-CpG cytosines located in short C stretches or right in front of CpGs are more difficult convert by bisulfite. We have adjusted our Discussion section to include this issue in the revised manuscript (line 478, page 19).

Minor:

Page 2, line 44. Reference "221" should be replaced by "1"

Page 3, line 70. Reference is lost

Page 9, line 223-226. Two first sentences of the Result section seem to be similar to each other and should be re-written

Page 13, line 335. Change 'S2A Fig.' to 'Fig. S2A'

We are grateful to the Reviewer for your comment. We have addressed the errors you noted, corrected the in-text citation, revised the first two sentences of the Results section (line 236, page 10) that were found to be similar. Additionally, we have adopted the recommended format for citing supporting information of “S Fig” instead of “Fig. S” as advised by PLoS ONE.

References

1. Lan VTT, Trang VL, Ngan NT, Son HV, Toan NL. An Internal Control for Evaluating Bisulfite Conversion in the Analysis of Short Stature Homeobox 2 Methylation in Lung Cancer. APJCP, 2019; 20(8), 2435-2443 Doi: 10.31557/APJCP.2019.20.8.2435

2. Morrison J, Koeman JM, Johnson BK, Foy KK, Beddows I, Zhou W, et al. Evaluation of whole-genome DNA methylation sequencing library preparation protocols. Epigenetics Chromatin. 2021; 14:28. doi: 10.1186/s13072-021-00401-y

3. Suzuki M, Liao W, Wos F, Johnston AD, DeGrazia J, Ishii J, et al. Whole-genome bisulfite sequencing with improved accuracy and cost. Genome Res. 2018;28(9):1364–1371. doi: 10.1101/gr.232587.117

4. Delaney C, Garg SK, Yung R. Analysis of DNA Methylation by Pyrosequencing. Methods Mol Biol. 2015;1343: 249–264. doi: 10.1007/978-1-4939-2963-4_19

---

## [Editor Report · Decision Letter 1]

26 Nov 2024

PONE-D-24-37696R1An appropriate DNA input for bisulfite conversion reveals LINE-1 and Alu hypermethylation in tissues and circulating cell-free DNA from cancersPLOS ONE

Dear Dr. Vo,

Thank you for submitting your manuscript to PLOS ONE. After careful consideration, we feel that it has merit but does not fully meet PLOS ONE’s publication criteria as it currently stands. Therefore, we invite you to submit a revised version of the manuscript that addresses the points raised during the review process.

We look forward to receiving your revised manuscript.

Kind regards,

Osman El-Maarri, Ph.D

Academic Editor

PLOS ONE
---

## [Author Response · Author response to Decision Letter 1]

27 Nov 2024

Dear Members of the Editorial Board,

We sincerely appreciate the guidance from the Editors and the comments from the Reviewers throughout the review process to improve our manuscript titled “An appropriate DNA input for bisulfite conversion reveals LINE-1 and Alu hypermethylation in tissues and circulating cell-free DNA from cancers” by Tran et al., submitted to PLoS ONE under the reference ID: PONE-D-24-37696R1. 

As per the journal requirements kindly provided by the Editor, we have now revised the manuscript to address these points. Modifications have been tracked in the revised manuscript and are detailed point by point in the following section.

To better adhere to the journal requirements, we have:

+ Ran our figures through the PACE tool for renaming and reformatting to meet technical requirements

+ Confirmed NO retracted paper was cited in our manuscript.

+ Outlined the specifics of the corrections for the five papers that have undergone author correction cited in our manuscript and the final state of these references in the revised manuscript. 

We are truly grateful for your time and efforts.

Best regards,

---

## [Editor Report · Decision Letter 2]

10 Dec 2024

An appropriate DNA input for bisulfite conversion reveals LINE-1 and Alu hypermethylation in tissues and circulating cell-free DNA from cancers

PONE-D-24-37696R2

Dear Dr. Vo,

We’re pleased to inform you that your manuscript has been judged scientifically suitable for publication and will be formally accepted for publication once it meets all outstanding technical requirements.

Kind regards,

Osman El-Maarri, Ph.D

Academic Editor

PLOS ONE
---

## [Editor Report · Acceptance letter]

17 Dec 2024

PONE-D-24-37696R2 

PLOS ONE

Dear Dr. Vo, 

I'm pleased to inform you that your manuscript has been deemed suitable for publication in PLOS ONE. Congratulations! Your manuscript is now being handed over to our production team.

Kind regards, 

on behalf of

Priv.-Doz. Dr. Osman El-Maarri 

Academic Editor

PLOS ONE